# Learning Translation Invariance in CNNs

**Valerio Biscione & Jeffrey Bowers**
Department of Psychology
University of Bristol
Bristol, BS81TU, UK
{valerio.biscione,j.bowers}@bristol.ac.uk

## Abstract

When seeing a new object, humans can immediately recognize it across different retinal locations: we say that the internal object representation is invariant to translation. It is commonly believed that Convolutional Neural Networks (CNNs) are architecturally invariant to translation thanks to the convolution and/or pooling operations they are endowed with. In fact, several works have found that these networks systematically fail to recognise new objects on untrained locations. In this work we show how, even though CNNs are not 'architecturally invariant' to translation, they can indeed 'learn' to be invariant to translation. We verified that this can be achieved by pretraining on ImageNet, and we found that it is also possible with much simpler datasets in which the items are fully translated across the input canvas. We investigated how this pretraining affected the internal network representations, finding that the invariance was almost always acquired, even though it was some times disrupted by further training due to catastrophic forgetting/interference. These experiments show how pretraining a network on an environment with the right 'latent' characteristics (a more naturalistic environment) can result in the network learning deep perceptual rules which would dramatically improve subsequent generalization.

## 1 Introduction

The equivalence of an objects across different viewpoints is considered a fundamental capacity of human vision recognition (Hummel, 2002). This is mediated by the inferior temporal cortex, which appears to provide the bases for scale, translation, and rotation invariance (Tanaka, 1996; O'Reilly & Munakata, 2019). Taking inspiration from biological models (LeCun et al., 1998), Artificial Neural Networks have been endowed with convolution and pooling operations (LeCun et al., 1998, 1990). It is often claimed that Convolutional Neural Networks (CNNs) are less susceptible to irrelevant sources of variation such as image translation, scaling, and other small deformations (Gens & Domingos, 2014; Xu et al., 2014; LeCun & Bengio, 1995; Fukushima, 1980). While it is difficult to overstate the importance of convolution and pooling operations in deep learning, their ability to make a network invariant to image transformations has been overestimated: for example, Gong et al. (2014) showed that CNNs achieve neither rotation nor scale invariance. As for 'online' translation invariance (the ability to identify an object at multiple locations after seeing it at one location), which is the focus of this work, multiple studies have reported highly limited online translation invariance (Kauderer-Abrams, 2017; Gong et al., 2014; Azulay & Weiss, 2019; Chen et al., 2017; Blything et al., 2020). This is generally measured by training a network on images placed on a certain location (generally the center of a canvas), and then testing the same images placed on untrained location.

The misconception about the ability of CNNs to be invariant to translation is due to the confusion between invariance and equivariance: it is commonly assumed that CNNs are 'architecturally'

Preprint. Under review at the 2nd Shared Visual Representations in Human and Machine Intelligence (SVRHM) Workshop at NeurIPS 2020.

invariant to translation (that is, the invariance is built in the architecture through pooling and/or convolution). For example: "Most deep learning networks make heavy use of a technique called convolution (LeCun, 1989), which constrains the neural connections in the network such that they innately capture a property known as translational invariance. This is essentially the idea that an object can slide around an image while maintaining its identity" (Marcus, 2018), see also LeCun & Bengio (1995); Gens & Domingos (2014); Xu et al. (2014); Marcos et al. (2016) for similar statements.

In fact, the convolution operation is translationally equivariant, not invariant, meaning that a transformation applied to the input is transferred to the output (Lenc & Vedaldi, 2019). Moreover, perfect equivariance can be lost in the convolutional layers due to using stride (Azulay & Weiss, 2019; Zhang, 2019) Therefore, overall, most modern CNNs are neither architecturally invariant nor perfectly equivariant to translation.

In contrast with most of the previous works finding a lack of translation invariance in CNNs, two recent studies have obtained a high degree of online translation invariance: Han et al. (2020) on a Korean Characters recognition task, and Blything et al. (2020) on a shape recognition task. Blything et al. (2020) firstly explained these incoherent results, finding that what would endow a network with translation invariance was pretraining on ImageNet: whereas such network would perform highly accurately on untrained location, a vanilla network (non-pretrained) would accurately perform only on the trained location, and at chance elsewhere. This hints to the fact that translationally invariant representations do not need to be built inside the network architecture, but can be learned. In the current work, we further explore this idea.

## 2 Current Work

In this work we focus on 'online' translation invariance on a classic CNN, using VGG16 (Simonyan & Zisserman, 2014) as a typical convolutional network. We show how, even though classic CNNs are not 'architectural' invariance, they can 'learn' to be invariant to translation by extracting latent features of their visual environment (the dataset).

We trained on environments in which the key characteristic was that items' categories were independent on their position (for CNNs learning the categories based on their position, see Semih Kayhan & van Gemert 2020). Our main contribution is finding that by pretraining on such environments, CNNs would indeed learn to be invariant to translation.

Why is this important? First, since CNNs have been recently suggested as a model for the human brain (Richards et al., 2019; Ma & Peters, 2020; Kriegeskorte, 2015; Zhuang et al., 2020), it is important to understand if and how they can learn fundamental perceptual properties of human vision, of which invariance to translation is one (Blything et al., 2020; Bowers et al., 2016; Koffka, 2013). Second, a network that learns deep characteristic of its visual environment such as being invariant to translation, rotation, etc., is able to accelerate subsequent training, and accordingly, it is important to understand the conditions that foster invariance.

We expand Blything et al. (2020) results on the ability of a CNN pretrained on ImageNet to show translation invariance, by testing this hypothesis on a wider variety of datasets (Section 2.2). We then show that it is possible to obtain similar results using much simpler artificial datasets in which objects were fully-translated across the canvas, but with some limitations due to the difference between the pretraining and the fine-tuning datasets (Section 2.3).This is likely due to catastrophic forgetting/interference, as shown in the internal representation analysis in Section 2.4.

### 2.1 Datasets

We used six datasets spanning a high range of complexity. From the more complex to the less complex[1], we used: EMNIST (Cohen et al., 2017); FashionMNIST (FMNIST), from Xiao et al. (2017), Kuzushiji MNIST (KMNIST), from Clanuwat et al. (2018); MNIST, from (LeCun et al., 1998); and two versions of the Leek dataset used in Blything et al.: one contained only 10 images

---

[1]For EMNIST, FashionMNIST, KMNIST and MNIST we based our judgment of complexity on the benchamrk in `https://paperswithcode.com/task/image-classification`. We deemed Leek10 and Leek2 as the easiest to classify due to the lack of intra-class variability, as each class is composed of just one image

instead of 24 of the original dataset (Leek10). The other contained only two images from the original dataset (Leek2), disjointed from the images used for Leek10. Representative examples (and, for Leek10 and Leek2, the entire datasets) are shown in Figure 1B. We did not apply any data-augmentation to the datasets (apart translating the items on the canvas, as explained below).

## 2.2 Experiment 1: Pretraining on ImageNet

The experimental design is shown in Figure 1A: we tested a VGG16 network either pretrained on ImageNet or not pretrained (vanilla). Both networks were then re-trained on a 1-location dataset, that is, a datasets where items from all classes were presented only on one location. The network that was pretrained on ImageNet would therefore use the learned parameters as initialization for the new training with the 1-location dataset (this is commonly referred as fine-tuning, Girshick et al. 2014), whereas the vanilla network would start from scratch with Kaiming Initialization (He et al., 2016). We used Adam optimizer with a fixed learning rate of 0,001. The 1-location dataset used items from the six datasets described in Section 2.1, but with each item resized to $50 \times 50$ pixels and placed on the leftmost-centered location on a black canvas $224 \times 224$, and thus no translation was used for this dataset. Both networks were then tested on the same items placed across the whole canvas. Therefore, the networks were tested on their ability to recognise trained objects on unseen locations ('online' translation invariance). We repeated each condition 5 times.

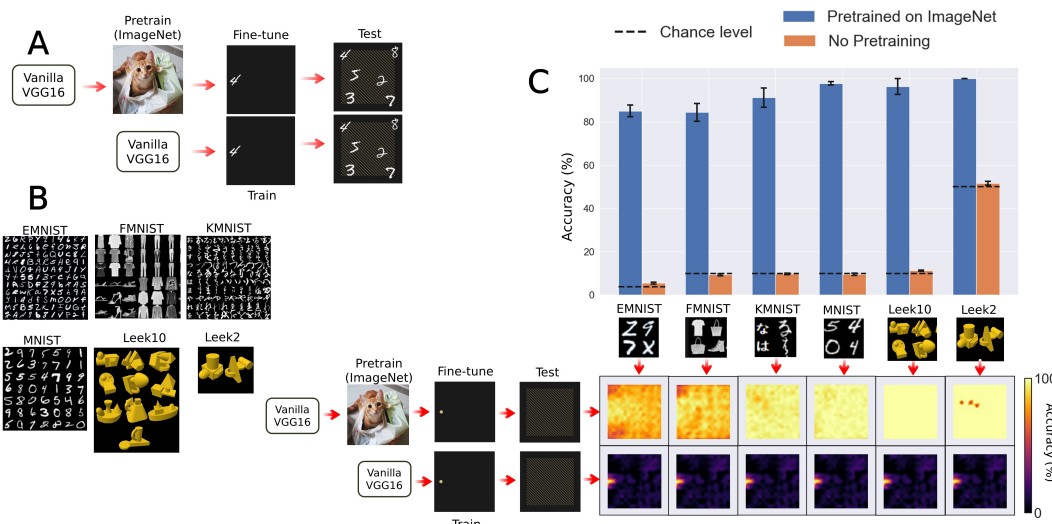

Figure 1: A. Representation of the general experimental design used here and in following experiments. B. Samples of the six datasets used here. Leek10 and Leek2 are shown in their entirety. C. Results from experiment 1, see text for details

The network pretrained on ImageNet was able to recognize with high accuracy objects from the 1-location datasets when tested on unseen locations, whereas a vanilla network was not able to do so (Figure 1C, barplot). We can better understand the extent of networks' invariance to translation by plotting its accuracy across the whole canvas: we tested the network on items centered across a grid of $19 \times 19$ points equally distributed upon the canvas. By interpolating the results across the tested area we obtained a heatmap like those shown in Figure1C, bottom. We can see that in the case of the vanilla network (bottom row), for all datasets, high accuracy was achieved only at trained location (the leftmost-centered one). On the other hand, the pretrained model (upper row) was extremely accurate everywhere on the canvas.

## 2.3 Experiment 2: Beyond ImageNet

We hypothesised that the network pretrained on ImageNet learned translation invariance because of the data augmentation performed during pretraining that involved translating subsets of the input images across the visual field of the network. If this is correct, we could hypothetically make a

network learn to be invariant to translation with any simple datasets in which the items occur all across the canvas.

We pretrained a VGG16 network on fully-translated datasets: items from datasets in Section 2.1 were resized to $50 \times 50$ pixels and randomly placed anywhere across a $224 \times 224$ black canvas. Similarly to the previous experiment, we then fine-tuned the network on each 1-location dataset, and tested it on the same items, fully-translated (Figure 2A). Again, if the network has learned to be invariant to translation, it would be able to recognise objects everywhere on the canvas, without need to be trained on every location. The resulting accuracy (normalized so that 0 is chance level), averaged across the whole canvas and across five repetitions for each (pretrain, fine-tune) pair, are shown in Figure 2B. In most cases, the networks were able to correctly classify items from the 1-location datasets when seen in untrained locations However, this is not true for all combination of pretrained and fine-tuned datasets. We observed a pattern in which networks pretrained with a "complex" fully-translated dataset obtained high performances when fine-tuned on a 1-location "simple" datasets, but the opposite was not true (for example, pretraining on Leek2 resulted in poor performance when fine-tuned on any other dataset). We further explore this finding in the next section.

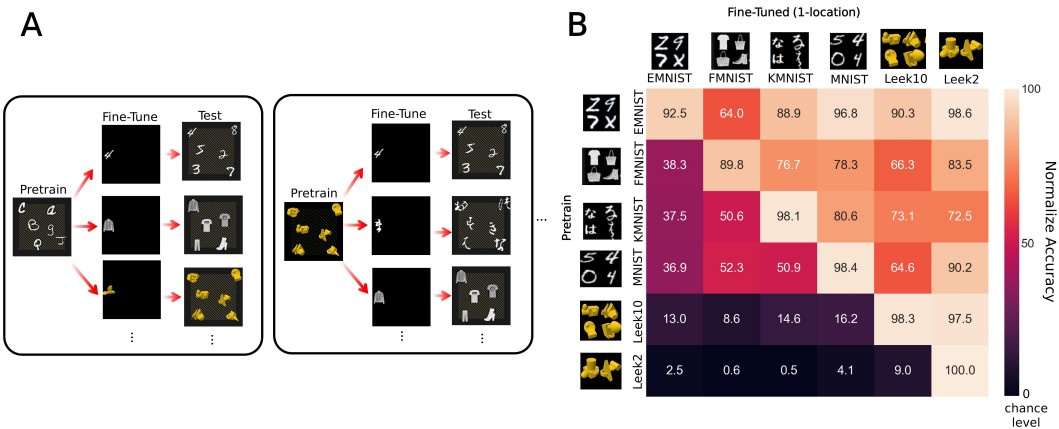

Figure 2: A. Experimental design for Experiment 2. B. Results in terms of normalized accuracy (0 is random chance, 100 is perfect performance).

## 2.4 Internal Representation with Cosine Similarity Analysis

The pattern found in the previous section can be explained by the phenomenon of interference and catastrophic forgetting (Furlanello et al., 2016; McCloskey & Cohen, 1989) in which training on new tasks degrades previously acquired capabilities. If this was true, it would mean that a network could learn to be invariant to translation *with any translated dataset*, regardless of its complexity, and this ability could be retained with techniques that prevent catastrophic forgetting (Beaulieu et al. 2020; Javed & White 2019; Li & Hoiem 2016).

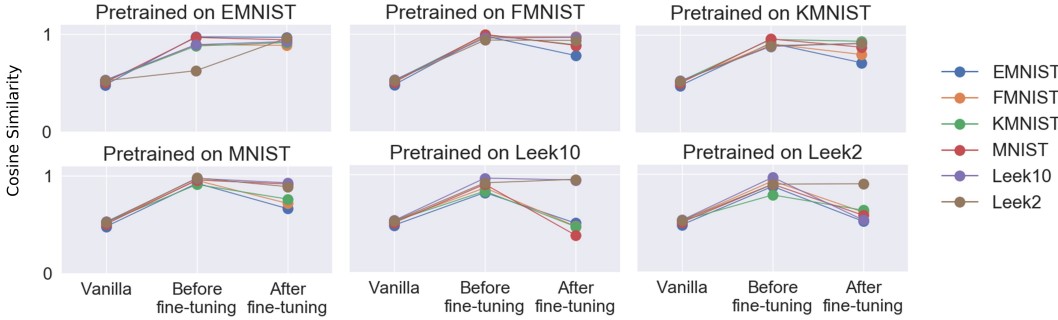

Figure 3: Cosine similarity analysis before and after 1-location fine-tuning, for each pretraining fully-translated dataset, showing that in almost every condition the network acquired translation invariance, but it was some times disrupted by fine-tuning.

We tested this hypothesis by computing the cosine similarity between the activations of the penultimate networks' layers when given as input an image at the leftmost-centered location and an images at different levels of displacement from that location. This metric indicates the degree to which translated objects share the same internal representation. We computed this metric for the networks trained in Section 2.3 at three stages: vanilla, before fine-tuning (that is, after pretraining) and after fine-tuning. For each pretrained network, we measured the cosine similarity for every dataset, which means measuring the similarity of internal representation for objects that, apart for the pretrained ones, the network had never been trained on. Results are shown in 3: a high degree of cosine similarity was observed before the fine-tuning, *for all datasets, and for networks pretrained on any dataset.* This similarity would then drop to an untrained level when the network was fine-tuned on more complex datasets.

## 3   DISCUSSION

Our experiments show that translation invariance can be learned in CNNs even though they lack built-in architectural invariance, and raise the possibility that a whole range of perceptual capabilities can be learned rather than built in the architecture. This work also suggest that, in certain cases, the architecture may be less important than the visual world the network is trained on.

Neural networks performance is often compared to human performance (Baker et al., 2018; Han et al., 2019; Srivastava et al., 2019; Ma & Peters, 2020). A fundamental feature of human perception is that it supports widespread generalization, including combinatorial generalization (e.g., identifying and understanding images composed of novel combination of known features). Current CNNs are poor at generalizing to novel environments (Geirhos et al., 2018), especially when combinatorial generalization is required (Vankov & Bowers, 2020; Hummel, 2013). Here we have shown that CNNs are able to extract latent principles of translation invariance from their visual world and to re-use them to identify novel stimuli with very different visual forms in untrained locations. An interesting question is the extent to which other forms of generalization and other fundamental principles of perception (e.g., Gestalt principles of organization, Koffka 2013) can be learned in standard CNNs trained on the appropriate datasets, and what sorts of generalization requires architectural innovations. Without the right training environment, it is not surprising that CNNs fail to capture the cognitive capacity of the human visual system (Funke et al., 2020), and the only way to address this fundamental question is to train models under more realistic conditions.

Although training on more naturalistic datasets may lead to better and more human-like forms of generalization, it is also worth highlighting how our artificial hand-crafted environments, such as our fully-translated datasets, could be used to train networks to acquire a particular perceptual regularity in spite of the architecture not directly supporting that representation. This may prove to be a useful technique for learning in more complex environments: instead of having a network learning all possible visual configurations (through data augmentation) of a given dataset, the network can be pretrained on extremely simple datasets that embed fundamental perceptual principles of the environment. When facing a complex dataset, the network only needs to learn the basic configuration of the new objects, and extrapolate the others through learned perceptual properties. This approach would clearly need to address the problem of catastrophic forgetting since, as we have seen, it can seriously impair generalization of learned perceptual rules when applied to new objects (Li & Hoiem, 2016; Furlanello et al., 2016).

## 4   CONCLUSION

We have shown that, even though standard CNNs are not architecturally invariant to translation, they can learn to be by training on a dataset that contains this regularity. We have also shown how such property is retained when fine-tuning on simpler dataset, but lost for more complex ones, and this appears to reflect the role of catastrophic interference in constraining translation invariance rather than a failure to learn invariance from these simple datasets. We suggest that by using more naturalistic environments, and an approach that would avoid catastrophic forgetting, we could unlock a greater capacity of generalization.

## Acknowledgments and Disclosure of Funding

This project has received funding from the European Research Council (ERC) under the European Union's Horizon 2020 research and innovation programme (grant agreement No 741134).

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

## A    Appendix

### A.1    Beyond ImageNet

In Section 2.3 we pretrained on the fully-translated version of each one of the the six datasets described in Section 2.1. We then fine-tuned the network on each 1-location dataset, and tested it on the same items, fully-translated. We repeated each training session 5 times. In Figure 2 we show the average performance across each repetition. Here we also show the standard deviation for each one of the 36 conditions (Figure 4).

### A.2    Sanity Check: Training on the whole canvas is not enough

It is conceivable that in the previous experiments, and in similarly designed experiments in the literature (Kauderer-Abrams, 2017; Gong et al., 2014; Chen et al., 2017; Blything et al., 2020), vanilla networks failed to show invariance to translation because they were tested on locations where they had not seen any items. In which case, pretrained networks succeeded not because they had acquired the deep property of translation invariance from the visual environment, but simply because they had been trained on the whole canvas. To test this hypothesis we separated the canvas in 9 equilateral areas ($58 \times 58$ pixels), and within each area, only 2 of the total classes were presented. The items

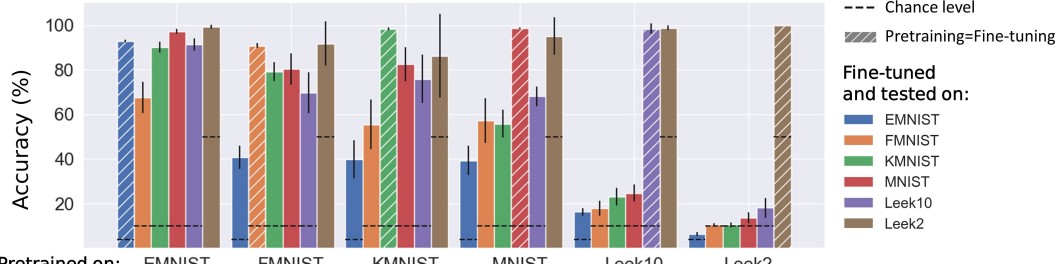

Figure 4: Average results across each repetition for Experiment 1. Error lines indicate one standard deviation. Hatches indicate the condition in which fine-tuning had the same items as pretraining (corresponding to the diagonal in Figure 2)

were randomly centered anywhere within their area (in such a way that part of the object could sometime slightly overlap another area, but the objects were never cropped). Therefore the objects were subjected to limited translation. Once trained with this setup, we fine-tuned the network on a 1-location dataset and tested the same dataset on the whole canvas (like in Section 2.2 and 2.3). We used the EMNIST dataset with the first 18 categories (EMNIST18) for pretraining, and MNIST for 1-location fine-tuning and fully-translated testing dataset, because they were the datasets that most consistently resulted in good translation invariance in the previous experiment. Results are shown in Figure 5A. We also tested the pretrained network on a fully-translated version of EMNIST18, that is, without class segregation, and without fine-tuning (Figure 5B). Even though the network was trained on items everywhere on the canvas, it did not acquire the ability to generalize on unseen locations with neither EMNIST18 nor newly trained objects (MNIST). This is a strong demonstration that the network needs to be trained on an environment where objects are translated across the whole canvas in order to learn to be invariant to translation, and that simply training on the whole canvas is not enough.

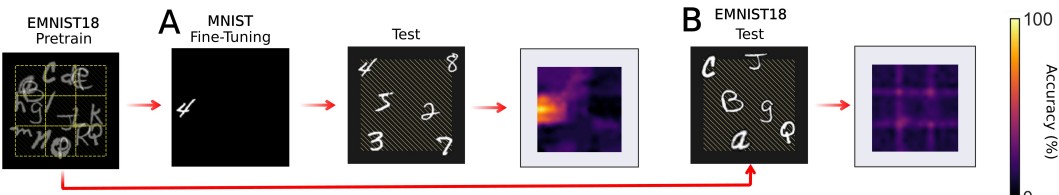

Figure 5: Experimental design for Experiment 3 and results. A. After fine-tuning on MNIST, the network did not generalize on untrained locations. B. When tested on the same dataset (EMNIST18), but without using classes segregation, letters were only recognised when presented on the area they were trained on, so mean accuracy was low. In both cases, the heatmaps are averaged across all classes.

