# OpenReview forum: "Learning Translation Invariance in CNNs"
_NeurIPS.cc/2020/Workshop/SVRHM — SVRHM@NeurIPS Poster_

### Official Review · AnonReviewer2 · 2020-10-29
**Interesting study of translation invariance**

**Rating:** 8
**Confidence:** 4

**Review:**

There is a lot of misleading statements made in the literature on the supposed translation invariance/equivariance of CNNs, and as such this paper is very welcome. The experiments both on pretraining a CNN with ImageNet as well as simpler datasets and testing for translation invariance are thorough enough to be convincing of the basic thesis - that while CNNs are not intrinsically invariant, they can learn invariant representations.

Here are some points that would make the paper stronger:
- It would be interesting to see how the forgetting and lack of performance scales with the complexity – we see that Leek2 and Leek10 generalize very poorly, but maybe one could subsample the other datasets and get some better understanding of the dependence on the size of the dataset
- What effect does the CNN architecture have here? As mentioned in the intro, stride has impact, but it would be nice to explore this in more detail.
- Some info on how fine-tuning was achieved while changing the number of classes (EMNIST, MNIST, Leek2 each having different #classes)

---

> ### Public Comment · ~Valerio_Biscione2 · 2020-11-30
> **Response to AnonReviewer2**
>
> We thank the reviewer for their comments.  We fully agree that testing different numbers of categories could be valuable in separating the factors involved in transferring translation invariance. In terms of architecture, we will consider including in an expanded version of this work tests on other architectures we have tried (VGG11, VGG13, ResNet18 with and without global average pooling), which all confirmed the results shown here.
>
> There are two common ways of achieving fine-tuning: replacing the last fully-connected layer with a new one with the correct number of output units, or adding an additional layer on top of the last one in the network. We used the first approach. We are aware that either approach may disrupt the learned translation invariance, and this is something that we will discuss in an expanded version of this work. The cosine similarity analysis, however, overcomes this problem, as it does not involve changing the network architecture.

---

### Official Review · AnonReviewer3 · 2020-10-30
**Analyses need more development**

**Rating:** 5
**Confidence:** 4

**Review:**

CNNs are not fully invariant to translation, as is often assumed. Recent work has found that pretraining on ImageNet allows the network to learn translation invariant representations on a transfer task. In this paper, the authors ask: _What is it about pretraining on ImageNet that endows CNNs with translation invariance?_

The authors first demonstrate the ImageNet pretraining result. The authors either pretrained a vanilla VGG16 network on ImageNet or began with an untrained model. They then fine-tuned or trained the network on a dataset in which objects appear in only one location. At test, the objects could appear in unseen locations. The accuracy of the network for different object translations is visualized in a heatmap. The network pretrained on ImageNet performs nearly perfectly across all 6 datasets considered, whereas the untrained network completely fails to generalize.

The authors hypothesize that pre-training on ImageNet facilitates translation invariance because during images appear in all possible positions on the canvas. They extend this hypothesis to ask whether pretraining on simple fully-translated datasets would allow the network to learn translation invariant representations in the transfer task. The authors find that while some fully-translated datasets allow the network to generalize, many do not. In particular, the MNIST variants transfer reasonably well between each other, but the Leek datasets completely fail to transfer to the MNIST variants.

The authors suggest that this is due to catastrophic forgetting. The authors analyze the cosine similarity of the representations of translated versions of the same object and find that all networks have high invariance before fine-tuning, which degrades severely after fine tuning.

I find the catastrophic forgetting argument unconvincing. Catastrophic forgetting is not an explanation of the cause of the failure to generalize in the previous task. Rather, it is a term that simply describes the phenomenon observed. It provides little explanatory power.

A more relevant hypothesis that is not considered in this work is that ImageNet is a rich dataset in which image features at many levels of abstraction appear in many positions on the canvas. Because of the rich structure in this dataset, the array of learned features include features that are relevant for solving other tasks, such as the MNIST variants. Because they appear in all locations, this gives the model an advantage at learning translation invariant representations. The structure in MNIST and Leek, on the other hand, is very narrow. It is unsurprising that the pre-training fails to generalize in some cases, as the models are fit to a very different set of features.

In short, it seems the simplest hypothesis is that training on a rich dataset in which 1) the feature set is sufficiently rich to generalize and 2) these features appear at all locations on the canvas allows a model to transfer useful translation invariant features to new tasks.

I would have liked to see this hypothesis considered and tested. In future work, it may also be fruitful to use feature visualization techniques to tease apart the learned structure of these models and pinpoint the causes of this phenomenon.

---

> ### Public Comment · ~Valerio_Biscione2 · 2020-11-30
> **Response to AnonReviewer3**
>
> We thank the reviewer for their comments.
>
> It is true that catastrophic forgetting describes a phenomenon, but the basis for the phenomenon is also well-understood, namely, new learning overwriting old learning by altering the weights of the network (McCloskey and Cohen, 1989). The important contribution here is that we are the first to show that catastrophic interference plays a role in constraining translation invariance.  That is, we found that the internal representations of novel objects (measured by cosine similarity) were translation invariant prior to training the network on novel objects at restricted locations on the canvas.  After training, however, both performance and the cosine similarity measure for some objects showed limited invariance.
>
> It would be certainly interesting to analyse what are the factors across pre-training and training datasets that contribute to the weights interference. For example, datasets similarity is most likely playing a role, but it is not the only factor: the network retains invariance to translation on Leek 10 or Leek2 when pre-trained on every other datasets, which have very dissimilar features. Similarly, all the datasets used here are very different to the images used when pre-training on ImageNet. Also relevant could be the number of classes the network is pre-trained on. We will consider expanding on the topic in a future version of this work.
>
> Reviewer 1 hypotheses that translation invariance may simply be due to the network being trained over all locations (as opposed to the network being trained on translated images) is a reasonable one. In fact, we tested this hypothesis in the Appendix A.2, in which we trained a network on EMNIST all across the canvas, but by separating the canvas in 9 areas and presenting only 2 classes within each area. In this condition, the network was not able to generalize translation across the whole canvas on a new dataset. We appreciate that the reviewer was not asked to read the Appendix and we will consider including this section in the main text when expanding this work.

---

### Decision · Program_Chairs · 2020-11-02

Accept (Poster)